# Histomorphological and Dynamical Changes in Female River Lampreys during Maturation under Controlled Conditions as a Part of Lamprey Restoration Programs

**DOI:** 10.3390/ani14172516

**Published:** 2024-08-29

**Authors:** Joanna Nowosad, Roman Kujawa, Shahid Sherzada, Dariusz Kucharczyk, Mateusz Mikiewicz, Katarzyna Dryl, Andrzej Kapusta, Joanna Łuczyńska, Hany M. R. Abdel-Latif

**Affiliations:** 1Department of Research and Development, Chemprof, 11-041 Olsztyn, Poland; 2Department of Ichthyology, Hydrobiology and Aquatic Ecology, National Inland Fisheries Research Institute,10-718 Olsztyn, Poland; andrzej.kapusta@infish.pl; 3Department of Ichthyology and Aquaculture, University of Warmia and Mazury in Olsztyn, 10-720 Olsztyn, Poland; reofish@uwm.edu.pl; 4Department of Zoology, Government College University, Lahore 54000, Pakistan; shahid.sherzada@gcu.edu.pk; 5Department of Pathological Anatomy, Faculty of Veterinary Medicine, University of Warmia and Mazury in Olsztyn, 10-719 Olsztyn, Poland; mateusz.mikiewicz@uwm.edu.pl; 6Department of Gamete and Embryo Biology, Institute of Animal Reproduction and Food Research, Polish Academy of Sciences, 10-243 Olsztyn, Poland; k.dryl@pan.olsztyn.pl; 7Department of Commodity and Food Analysis, University of Warmia and Mazury in Olsztyn, ul. Plac Cieszyński 1, 10-726 Olsztyn, Poland; jlucz@uwm.edu.pl; 8Department of Poultry and Fish Diseases, Faculty of Veterinary Medicine, Alexandria University, Alexandria 22758, Egypt

**Keywords:** Petromyzontidae, *Lampetra fluviatilis*, fecundity, gonadosomatic index, oocyte maturation

## Abstract

**Simple Summary:**

Lampreys are a group of about 40 species found all over the world. One representative of this group is the river lamprey, which spends its larvae life initially in fresh water and then in the sea. For spawning, which is the end of their last migration, they return to fresh water. The spawning migration period lasts many months, during which the lampreys do not eat and many changes occur in their bodies. However, knowledge of these processes and their dynamics has not been studied and described, and this is the basis for preparing programs for the restitution of lampreys and their artificial reproduction under controlled conditions. This paper describes the changes occurring during spawning migration in female river lampreys, including the development of ovaries, changes in the liver, and atrophy of the digestive system.

**Abstract:**

More than 40 species of lampreys (Petromyzontiformes) are known worldwide. Some of them are parasitic and feed on the blood of fish or other aquatic animals. Lampreys spawn once in their lifetime, after which they die. One of the representatives of the ichthyofauna of European rivers is the river lamprey, *Lampetra fluviatilis*. The river lamprey is now an endangered species due to loss and degradation of their habitats. The present study investigated gonadal development without hormonal stimulation in female river lampreys during puberty under controlled conditions for a period of seven months. Female river lampreys were kept in conditions that mimicked the natural environment. During the November–May period, samples were taken monthly to determine the extent of gonadal development and gastrointestinal and liver changes using histological examination. From the results obtained, the dynamical changes were determined for the following: gonadosomatic index (GSI; %), hepatosomatic index (his; %), and digestivesomatic index (DSI; %). With the gonadal development of female lampreys, an increase in GSI (7–23%; November–May) and a decrease in DSI (0.4–0.1%; November–May) histological changes were observed in the gonads (oocyte development), intestines (over time, decreased lipid vacuoles and enterocyte apoptosis), and in the liver (decreased lipid vacuoles and hepatocyte apoptosis over time) and in the digestive system resulting from its degradation. Also, it was observed that there was a change in the color of the liver to green in April. This study demonstrated the key role of the liver in the oocyte maturation process in this species.

## 1. Introduction

Lampreys (Petromyzontiformes) are primitive aquatic animals included in the jawless vertebrate group (agnathans) [1]. Lampreys have become one of the most important research models for understanding vertebrates and human disease [2,3]. Currently, about 40 species [4,5] of these animals are known. They inhabit both marine and freshwater. Within this group, there are parasitic and non-parasitic species [4]. Parasitic lampreys attach themselves to the bodies of aquatic animals, mainly fish, and feed on their blood or tissues [6]. These animals have a complex life cycle that includes a long period of freshwater larvae (ammocoete; 3–6 years), followed by metamorphosis into sexually immature juveniles and then mature adults that differ significantly in morphology and ecology from larvae [1,7,8]. Three lamprey species are an important part of commercial fisheries in the northern hemisphere: the sea lamprey (Petromyzon marinus) in the Iberian Peninsula and France, the European river lamprey (*Lampetra fluviatilis*) in the Baltic Sea countries and Russia, and the Pacific lamprey (*Entosphenus tridentatus*) in Russia [9]. One of the most common representatives of the family of lampreys of the European ichthyofauna is the river lamprey *Lampetra fluviatilis* [10].

Like the European eel, *Anguilla anguilla*, the river lamprey is a bi-environmental species with spawning migrations. The European eel is a catadromous species, residing in freshwater and migrating to the sea to spawn [11,12]. In contrast to the eel, the river lamprey is an anadromous species [13], which spends the period of its proper adult life in saline waters and then, at the age of five to six years, in autumn (October–November) begins its upstream spawning migration. The spawning migration lasts for several months, and spawning occurs in late spring [14]. After selecting a convenient site optimal for offspring development, females deposit eggs into a nest in a gravelly and sandy bottom [10,15,16,17]. Lamprey larvae remain in freshwater for up to several years buried in the muddy bottom of stagnant rivers, feeding on algae, protozoa, and plant debris [10,17,18]. After reaching only several centimeters, the larvae undergo transformation and swim to the seas [7]. While living in marine waters, the parasitic river lampreys feed on the blood of fish; often, these species are economically valuable, e.g., sea trout (*Salmo trutta m. trutta*), Atlantic salmon (*Salmo salar*), cod (*Gadus morhua*), herring (*Clupea harengus*), or smelt (*Osmerus eperlanus*) [10,19,20,21].

The digestive system of lampreys that undertake spawning migrations atrophies because of a long-term cessation of food intake (8–9 months) [6,22]. After spawning, following endocrine-mediated physiological changes, lampreys die [17,23,24]. Formerly, the lamprey was widespread through western Europe, but it is now regarded as a threatened species and receives conservation protection in Europe through the Bern Convention and the European Habitats Directive 92/43/EEC [25,26]. The existence of this species in Poland is restricted to the northwestern part of the country, especially in the Wieprza, Rega, and Parsęta rivers [19]. According to the threat category of the Polish Red Book of Animals, the river lamprey is a high-risk species at risk of extinction (VU). Many varied factors, including overfishing, water pollution, or river regulation, have contributed to the drastic decline of these animals [10,16,25,26,27]. In addition, hydrological developments in rivers make it difficult for lampreys to reach spawning grounds [28,29]. Undoubtedly, the reproductive efficiency of lampreys is influenced by the level of accumulated nutrients in tissues prior to the start of the reproductive migration. Lampreys do not take food during the reproductive migration, which lasts several months, or during the spawning act itself [23,24,30].

Besides the passive protection measures and procedures, active protection involving population enhancement through stocking is also important. However, to carry out this process, it is first necessary to learn the details of the reproductive biology of the species in question and then to develop the biotechnique of reproduction and the production of young for stocking material [6]. The process of learning about the reproductive biology can be carried out partly under controlled conditions, for example, when keeping fish and lampreys and carrying out basic research on the changes taking place in ovarian and testicular development and gamete maturation [23]. Equally important and especially useful when carrying out the long process of gonadal maturation is to check the changes occurring in the gastrointestinal tract (especially for species that no longer take food) and the changes occurring in the liver [24,30]. This organ is responsible, among others, for protein and fat metabolism and for transporting them from body tissues (mainly muscle) and allocating them to the development of gonads and gametes. It is impossible to accurately trace such a process under natural conditions. An additional difficulty is that these animals are constantly on the move during reproductive migration, and it is not easy to obtain specimens for study [6]. Therefore, where possible, such studies must be conducted under controlled conditions, where environmental conditions such as salinity, photoperiod, temperature, and water current mimic natural conditions [18,20]. Interest in changes occurring during lamprey spawning migration has been reinforced by the growing need to restore native populations of this species [17,31]. The artificial production of lampreys is a method that can be used for restoring some imperiled populations [32]. Up to now, no studies have been conducted to elucidate the changes that may occur in the bodies of female river lampreys during spawning migration. Therefore, the aim of the present study was to trace changes in the bodies of female lampreys during spawning migration, considering histomorphological changes in the digestive system, liver, and gonads. The findings of this study will provide a comprehensive outline about the dynamical changes that occur during the spawning migration of this species. 

## 2. Materials and Methods

### 2.1. Ethics

At this time, experiments with river lamprey do not need any Animal Care Commission approval, but permission from the General Director for Environmental Protection must be granted for obtaining adult lampreys from the environment (Decision number DOPOZ.6401.10.3.2013.ls). The research was conducted on a protected species in Poland after obtaining the legally required decision and in accordance with the welfare of animals and good aquaculture practice.

### 2.2. River Lamprey Origin

River lampreys were harvested in autumn (October) during spawning migrations near Frombork town (coordinates: 54°21′ N; 19°40′ E) in the Vistula Lagoon (Vistula delta region—northern Poland) using tunnel nets. The river lampreys were carefully removed from the nets and placed in containers with aerated water. The experimental animals were then transported in plastic bags with oxygen to a hatchery at the Department of Aquaculture and Fisheries of the University of Warmia and Mazury in Olsztyn.

### 2.3. Broodstock Management

In the hatchery, lampreys were placed in spawner tanks with a volume of 1 m^3^ each [33]. All tanks were equipped with an independent oxygen supply, temperature and photoperiod control system, biological filters, and UV (ultraviolet) lamps. The water temperature was maintained with that found in the Vistula Lagoon during sampling (i.e., 8 °C, both in October). From March, the water temperature began to be gradually raised to 15 °C (May). The appropriate water movement imitating a river current and proper water variable were ensured using an external EHEIM CLASSIC 2260 filter (1500 L; Germany) as well as by intensive aeration using a pipe diffuser. Low water temperature in the pool was ensured through the use of an automatic water-cooling system. The values for water variables were recorded using a central OxyGuard Pacific system. Walls the tanks was dark green. Because lampreys do not feed during spawning migration, they were not fed during this study. The tanks were tightly covered to prevent lampreys from escaping [19,20,21].

### 2.4. Sex Determination

The river lampreys were handled with the utmost care. Sex determination was impossible in adults collected in the autumn, although lampreys had distinguishing sex characteristics in the spring immediately before spawning. In lampreys, because sexual dimorphism is minimally expressed outside the breeding season, the sex determination was by use of the ultrasonography (USG) method described by Kujawa et al. [21]. Before sex distinction, the animals were anesthetized with an MS-222 (Finquel, Los Angeles, CA, USA) solution (dose 100 g/m^3^). Females were placed in separate tanks (n = 42).

### 2.5. Sampling and Morphological Indexes

Samples were collected monthly (from six individuals, n = 6) for a period of seven months (November to May). Females were trapped from the tank and euthanized with an overdose of MS-222 [34]. Before sampling, lampreys were weighed individually (±0.01 g) and measured (±1.0 mm), and organs (liver, gonads, intestine) were sampled and then dissected from all specimens for further analyses. The collected organs were weighed on a precision balance (KERN & Sohn GmbH) with an accuracy of ±1 mg. The samples were frozen at −80 °C until use (Sanyo, MDF-U32V).

The following indices were calculated: 

Gonadosomatic index ((GSI; %) = GW × 100/BW, where GW: gonad weight (g), BW: body weight (g));Total fecundity (F = EN × GW, where EN: number of eggs in 1 g (pieces g), GW: gonad weight (g));Hepatosomatic index ((HSI; %) = HW × 100/BW, where HW: liver weight (g), BW: body weight (g));Digestivesomatic index (DIS; %) = (DW BW − 1) 100%, where DW: digestive tract weight (g), BW: body weight (g).

### 2.6. Histological Analysis

The tissue fragments from gonads, livers, and middle intestines were fixed in Bouin’s solution. The fixed tissues were added into biopsy cartridges and then into a tissue processor (LEICA TD 1020, Wetzlar, Germany) for 21 h, where the samples were washed in ethanol at increasing concentrations (75, 80, 90, and 95%) of acetone, xylene, and liquid paraffin at 54 °C. The obtained material was sealed in paraffin blocks and sliced in a rotating microtome (LEICA RM 2155, Wetzlar, Germany) into 6–7 μm thick sequences. The paraffin sections were added onto protein-covered slides. The prepared slides were stained with Hemotoxin and Eosin (E&H stain; [35]) and periodic acid–Schiff according to McManus [PAS] [36]. Subsequently, the stained preparations were sealed with coverslips and histokitt (Glaswarenfabrik Karl Hecht GmbH & Co KG, Sondheim vor der Rhön, Germany). The histological slides were evaluated under a light microscope (BX63, Olympus, Tokyo, Japan) using CellSense (Olympus, Tokyo, Japan) software (V2.2).

### 2.7. Statistical Analysis

Data are presented as means ± standard deviation (mean ± SD). Normality and homogeneity of variances were performed using the Shapiro–Wilk and Leven tests. One-way ANOVA was carried out and Tukey’s test was used as a post hoc procedure, whereas *p* < 0.05 was used to indicate statistically significant differences between groups of females. Statistical analyses were performed using Statistica 13.1 PL software (TIBCO Software Inc.; Palo Alto, CA, USA).

## 3. Results

### 3.1. Gonadal Development

The maturation of female lampreys was gradual. During spontaneous gonad maturation in female river lamprey, there was a noticeable increase in the GSI index from 7 (November) to over 23% (May). The biggest significant increases in this parameter were observed in March (*p* > 0.05; Figure 1A). Despite the gradual increase in GSI index, a gradual decrease in female weight was noticed. From November to May, female lampreys lost approximately 30% of their initial body weight (Table 1). An increase in oocyte diameter and development was observed monthly. In November, some oocytes were over 0.5 mm in diameter (Figure 1B). It was noted that most of the oocytes were in the late vitellogenesis stage. This oocyte size class was also found in May, just before the onset of ovulation. Fat deposits were visible in the cytoplasm of the gonads (Figure 2A). In April and May, females had more than 60% of oocytes over 0.7 mm in diameter. Just before spawning, only 10% of oocytes had diameters above 0.9 mm (Figure 1B and Figure 2B). The total fecundity was 29,954 ± 3065 (rank: 22,299–42,019) eggs per individual.

### 3.2. Liver

Analysis of liver index values in female lampreys showed that from November to March, the HSI value was stable at around 1.3% (Figure 3). The highest HSI values of more than 2.5% were found in April. In May, a decrease in HSI values was again observed (Figure 3).

Of interest, macroscopic observations indicated a change in the liver color of lampreys, whereas, during the spring period (April–May), the liver turned green from brownish-brown (Figure 4A,B). Moreover, the liver in November had a normal structure, and one large lipid vacuole could be observed in numerous hepatocytes (Figure 5A). In addition, no glycogen was stored in the hepatocytes. In February, hepatocytes without fatty vacuoles with slightly granular cytoplasm were observed (Figure 5B). In March, single hepatocytes undergoing apoptosis were seen. In April, clusters of hepatocytes with vacuoles filled with slightly acidophilic protein fluid were present (Figure 5C). In May, increased clusters of hepatocytes without fatty vacuoles and covered by necrotic lesions were observed (Figure 5D). PAS-stained sections of the livers of females during May showed polysaccharides between the hepatocyte lobules and around the blood vessels with the presence of small glycogen granules in individual hepatocytes (Figure 5E).

### 3.3. Digestive System

In November, at the beginning of the migration of female river lampreys, the digestive tract accounted about 0.51% of their body weight (DSI). The DSI value in November was significantly bigger than in the following months (*p* < 0.05). A marked change in this parameter was observed between November and January. By January, the DSI value had decreased to 0.17% (Figure 6). 

Initially, the cross-sectional diameter of the digestive tract was approximately 2.5 mm (Figure 7). Before spawning (May), the digestive tract of female lampreys amounted to only 0.06–0.08% of body weight (Figure 6). At which time, the diameter of the intestine was about 0.65 mm (Figure 7). Histological analysis showed that in lampreys starting spawning migration, the intestine still had fully developed intestinal villi, clearly visible enterocytes without histopathological (degenerative) changes. In the case of females, many advanced histopathological changes were found in sections taken before spawning itself. The intestinal villi were disrupted, the intestinal mucosa was compromised, and the spiral valve of the intestine was severely damaged. The smooth muscle of the intestine, the intestinal blood vessels, and the shaft of the intestinal villi were atrophied.

## 4. Discussion

Reproduction is one of the most important vital functions ensuring the continuity and survival of species. Through adverse changes in the natural aquatic environment, hydraulic engineering structures, and water pollution, spawning grounds are often destroyed, or it is impossible for migratory fish to reach their spawning grounds. The result is depletion in numbers and even disappearance of species and/or individual populations [17,26,28,29]. Understanding the reproductive biology of a species is the basis for developing protocols for use under controlled conditions to produce stocking material and restore endangered fish stocks [16,20,21]. In the present study, the dynamics of the development of gonads, oocytes, and changes in liver size and intestines of female river lampreys were observed during the seven months preceding the spawning of these animals.

### 4.1. Gonads and Gametes

In lampreys, ovarian differentiation occurs at the larval stage [37]. River lamprey larvae (ammocoetes) at ages 3 and 4, about 10 cm in length, have oocytes at 0.08 ± 0.003 mm, with an average of 1000—26,000 oocytes [38]. In metamorphosed lampreys (body length 10.1 cm), the mean oocyte diameter was 0.1 mm [38]. After metamorphosis, the feeding river lamprey remains sexually immature. Vitellogenesis does not occur until the end of the parasitic lifestyle and progresses during the onset of spawning migration [6]. In the present study, gradual development of gonads and oocytes was observed from November to March, indicating that slow but balanced ovarian development was taking place under suitable environmental conditions. During spontaneous gonad maturation in female river lamprey, there was a noticeable increase in the GSI index from 7% to about 23% in November and May, respectively. Similar GSI results for this species in May, just before spawning (22.8%) in females taken from the wild, were previously determined by Dziewulska and Domagała [39]. This may indicate enhanced environmental conditions during the maturation of these animals under controlled conditions. For example, the GSI in females of the largest of the sea lamprey (*Petromyzon marinus*) caught in upstream captures in Minho River (Spain) in May was 14.8% [40]. However, a greater GSI observed in in sea lampreys caught in the Cheboygan River (Michigan, USA) was 21 ± 3.0% (BW = 250 ± 47 g; [41]).

While monitoring the development of gonads in the river lampreys in this study, an increase in the diameter of oocytes and their synchronous development were observed in individual months. Interestingly, the cell nucleus was already moved towards the outer wall of the oocytes. In bony fish, the nucleus remains inside the oocyte for a long time, and only before ovulation does it begin to migrate from the center of the oocyte [42]. An earlier report by Kucharyaveyi et al. [43] found that the diameter of river lamprey oocytes was 0.8–1.0 mm. In this study, in November, some oocytes had a diameter above 0.5 mm. In April and May, over 60% of oocytes had a diameter above 0.7 mm (<0.7–1.0 mm). Similar differences in the size of oocytes were found in the river lampreys obtained from the natural environment by Dziewulska and Domagała [39]. It is assumed that not all oocytes reach maturity. Thus, in lampreys, despite spawning only once in their lives, not all eggs reach maturity. This should be related to the amount of stored reserves in their body and their transport from the muscles to the gonads and then to the gametes during spawning migration. When the amount of substance is not enough or when gamete maturation occurs too quickly (e.g., due to a rapid increase in water temperature due to climatic anomalies), not all egg cells mature and are released during spawning.

According to the literature, fecundity in this species may range from 7500 to 28,000 [10]. The fecundity of female river lampreys in the present study was 29,954 ± 3065 eggs/female (rank: 22,299–42,019). For example, the fecundity of the sea lamprey ranges from 124,000 to 260,000 eggs/female [10]. The fecundity of sea lampreys in Lake Champlain in relation to the mean wet weight 173.8 g was 67,660 ± 6870 eggs/female [44] and in Lake Superior from 68,584 to 80,228 eggs [45]. Therefore, it is assumed that the total fertility of the river lampreys is not high compared to other species of this family.

### 4.2. Liver

In the present study, changes in the size and morphology of the liver were observed, which may indicate intense processes occurring in this organ during the maturation of female lampreys. The highest HSI value was observed in April (2.5%), a month before spawning. A higher HSI value during spawning was also observed in sea lampreys [40]. In migrating sea lampreys, the HSI value was 1.65%, while in those spawning, it was 1.74% [40]. One of the most essential functions of the liver is the synthesis and secretion of bile [46,47]. Bile acids and salts participate in the process of digestion and absorption of lipids in the small intestine. In humans, over 90% of bile acids in the intestine are absorbed and transported back to the liver through the blood [46]. In adult lampreys, atresia of the bile ducts leads to the disappearance of the gallbladder [47,48]. Yeh et al. [47] discovered that sea lampreys adapt to atresia of the bile ducts through a unique mechanism of de novo synthesis and secretion of bile salts in the intestines while simultaneously reducing the synthesis of bile salts in the liver.

In April of this study, green liver was observed in some females. At this time, a slightly acidic protein fluid was observed histologically in the liver, which may be related to this fact. The green color is the result of atresia of the bile ducts and transformations occurred in the liver. This may happen due to the increase in estradiol concentration in the lamprey blood and the accumulation of bile acids and pigments (such as bilirubin, biliverdin, and hemosiderin) in hepatocytes because of the degradation of erythrocyte hemoglobin [23]. Green liver syndrome has also been observed in other lampreys [49] as well as in farmed fish, such as yellowtail, *Seriola quinqueradiata* [50], and red sea bream, *Pagrus major* [51], fed with a low-fish meal diet. The authors explained this fact using a deficiency of taurine in the diet [50,51]. They elucidated that the occurrence of green liver occurred due to hemolytic biliverdin overproduction caused by dietary taurine deficiency [52]. In the present study, at the beginning of migration (November–October), histological examination of the livers of females revealed hepatocytes filled with lipid vacuoles. In spring, active fat lipolysis and necrotic changes in hepatocytes were observed. According to Savina et al. [52], during the spawning migration of river lampreys, there is a significant decrease in liver function in winter (January–February). This may be associated with the lack of food during the migration of lampreys (as they do not feed) and depletion of energy reserves. The decreased water temperature during this period also influences this process. According to Savina et al. [53], during anadromous migration, glycogen is not stored in the lamprey liver, which was also confirmed in this study. Therefore, metabolic changes in the liver during this period rely on the oxidation of fatty acids, and glycolytic ATP production is minimal [54,55]. However, in this study, in May, histological examination of the livers of females revealed polysaccharides between hepatocyte cords and around blood vessels and fine glycogen granules in individual hepatocytes. It has been demonstrated that mature sea lampreys are able to quickly recover after disturbances from the metabolic and acid–base imbalances incurred due to intense exertion. Within four hours, phosphocreatine regeneration, muscle adenylate, lactate clearance, and glycogen replenishment occur [6]. The glycogen observed in females in May in this study is accumulated for the final effort associated with spawning itself. As a result of the increase in hormone levels and activity, including estradiol in females, rapid transformations occur, accompanied by a significant increase in liver functions, which may be associated with the statistically significant increase in HSI values observed in April (2.5%) in this study.

Like other vertebrates, the liver plays a key role in gonadal maturation. In this organ, estrogen induces the synthesis of vitellogenin, the main phospholipid-glycoprotein precursor of egg yolk proteins, which are delivered by the bloodstream to the developing oocyte [42,55,56]. As gonads mature (oocyte growth), liver cells (hepatocytes) age and undergo programmed cell death (apoptosis). The significant increase in HSI values observed in April may be linked to the occurrence of a change in liver color to green and the observation of slightly acidic fluid in the histological image of liver hepatocytes.

### 4.3. The Digestive System

With the development of female gonads, the degradation of the digestive system was observed. In November, the intestines of lampreys still had fully developed intestinal villi and clearly visible enterocytes without significant degenerative changes. However, in lampreys just before spawning (April–May), numerous histopathological changes were observed. Intestinal villi were torn apart, the intestinal mucosa was damaged, and the spiral valve of the intestine was severely compromised. Smooth muscles of the intestine, intestinal blood vessels, and the core of intestinal villi were in a state of atrophy. Similar changes were observed, for example, in the Japanese lamprey (*Lampetra japonica*) [57], Caspian lamprey, *Caspiomyzon wagneri* [58], and sea lamprey [59]. The degradation of the digestive system was also observed in fish undergoing catadromous migrations, which spawn only once in their lifetime, such as the European eel (*Anguilla anguilla*) and American eel (*Anguilla rostrata*) [60]. With increasing maturity in the above-mentioned species, a reduction in intestinal size, a decrease in the thickness of the intestinal muscle layer, and a decrease in the number of intestinal villi and mucous cells were observed. In fully mature eels, advanced degeneration of the intestinal muscle layer was demonstrated [60]. Authors attributed the degenerative changes in eels to elevated cortisol levels [60].

## 5. Conclusions

During the 6–7-month spawning migration of lampreys, gonad development occurs. As shown in this study, lampreys derive all their energy for oocyte development from fat stored in their tissues such as the liver, intestine, and muscles. In their natural environment, to conserve energy during migration, lampreys avoid fast currents by resting on the bottom (e.g., stones) [6,14]. As gonads mature and oocytes grow, a simultaneous decrease in the amount of fat vacuoles in hepatocytes and aging of liver cells was observed, followed by destructive changes in this organ. Presumably, lipids needed for gonad development also came from the intestine. During the maturation of females, the degradation of this organ and a decrease in the amount of lipid vacuoles were also observed. Mature female lampreys do not possess gallbladder or bile ducts, resulting in the liver turning green in the final phase of maturation. Presumably, these animals have developed an adaptive mechanism to such a condition: on the one hand, to separate lipid metabolism during spawning fasting, and on the other, to utilize energy in the form of lipids for gonadal development. In maturing female river lampreys, the liver plays the most significant role in reproduction. This organ, by participating in both vitellogenin synthesis and lipid reallocation to maintain vital functions, conducts reproductive migration and completes reproductive development. The reproductive process of river lampreys is a key element of their life cycle and plays an essential role in maintaining the population of this species. Many aquatic animals, including river lampreys, are currently facing various threats such as habitat loss and environmental changes, which may negatively impact their ability to reproduce effectively. Therefore, the protection and restoration of natural spawning habitats are crucial for the conservation of river lamprey populations.

## Figures and Tables

**Figure 1 animals-14-02516-f001:**
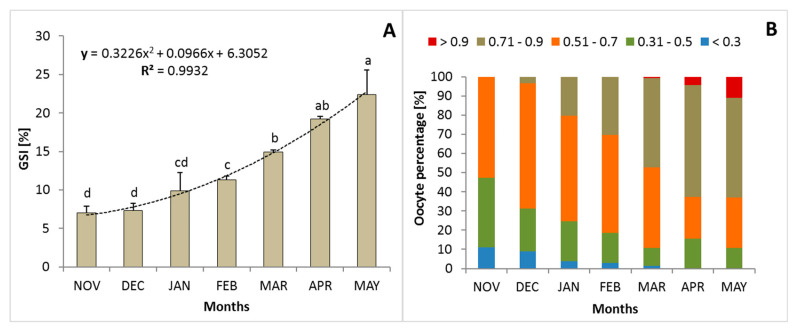
Dynamic changes in gonadosomatic index values (GSI values; %) (**A**) and oocyte diameter class (mm) (**B**) in female river lamprey, *Lampetra fluviatilis,* between November and May (n = 42), held under controlled conditions mimicking natural conditions (water temperature and photoperiod) during spawning migration. Columns in Figure 1A are marked with different letters to indicate statistically significant differences (*p* < 0.05; Tukey’s test).

**Figure 2 animals-14-02516-f002:**
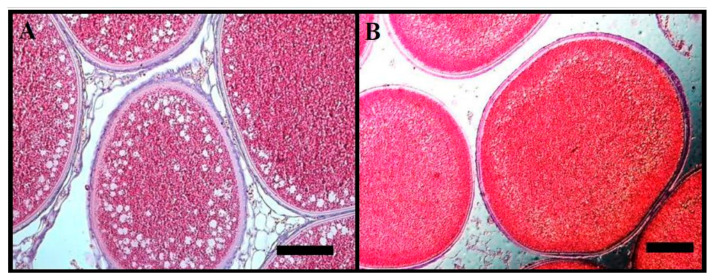
Photomicrographs (H&E) indicate the histological structure of oocytes collected from female river lamprey, *Lampetra fluviatilis,* from November (**A**) and May (**B**) (bar = 150 µm), held under controlled conditions mimicking natural conditions (water temperature and photoperiod) during spawning migration.

**Figure 3 animals-14-02516-f003:**
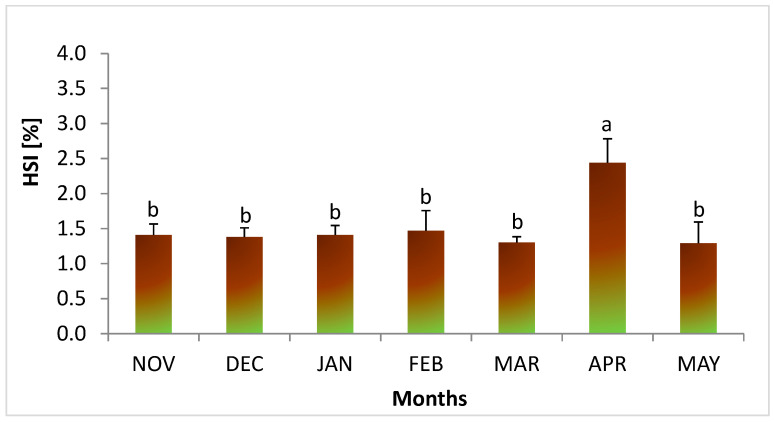
Dynamical changes in hepatosomatic index (HSI; %) values during the maturation of female river lamprey, *Lampetra fluviatilis* (n = 42), held under controlled conditions mimicking natural conditions (water temperature and photoperiod) during spawning migration. Columns are marked with different letters to indicate statistically significant differences (*p* < 0.05; Tukey’s test).

**Figure 4 animals-14-02516-f004:**
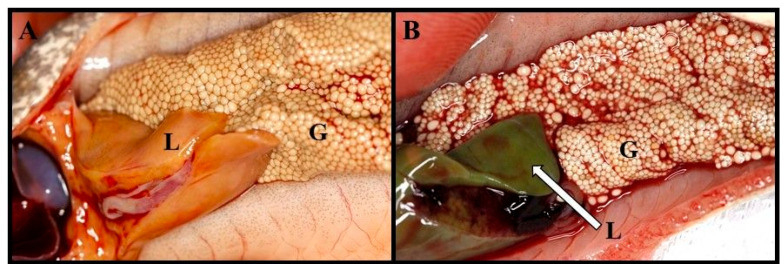
Changes observed in liver color from brownish-brown ((**A**) March) to green ((**B**) May) in female river lamprey, *Lampetra fluviatilis,* during puberty, held under controlled conditions mimicking natural conditions (water temperature and photoperiod) during spawning migration. L—liver, G—gonads.

**Figure 5 animals-14-02516-f005:**
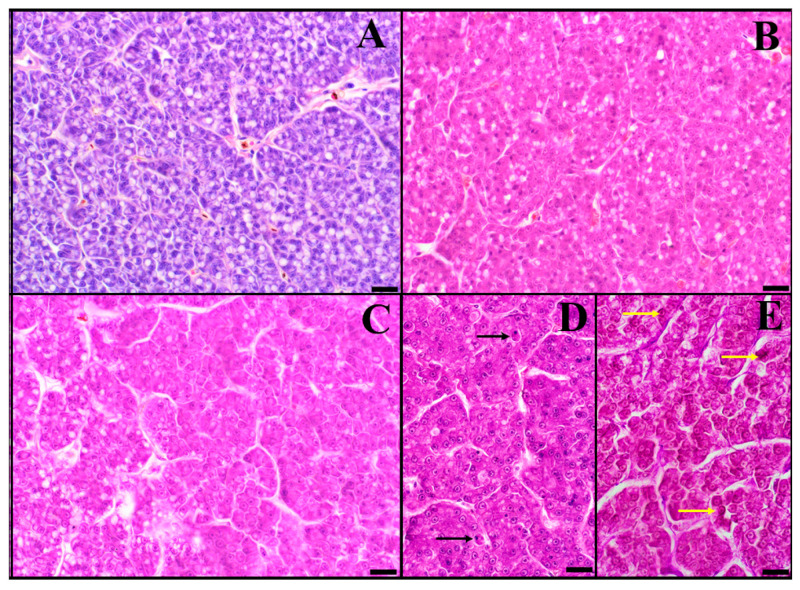
Photomicrographs ((**A**–**D**)—H&E; (**E**)—PAS) indicate the histological structure of livers collected from female river lamprey, *Lampetra fluviatilis,* from November (**A**), February (**B**), April (**C**), and May (**D**,**E**) (bar = 20 μm), held under controlled conditions mimicking natural conditions (water temperature and photoperiod) during spawning migration. The black arrow indicates cell apoptosis and the yellow arrow indicates glycogen granules.

**Figure 6 animals-14-02516-f006:**
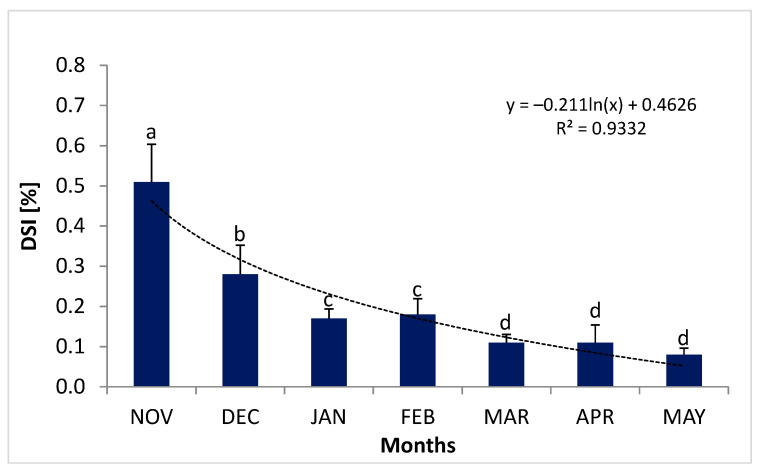
Dynamical changes in digestive somatic index (DSI; %) values during the maturation of female river lamprey, *Lampetra fluviatilis* (n = 42), held under controlled conditions mimicking natural conditions (water temperature and photoperiod) during spawning migration. Columns are marked with different to letters indicate statistically significant differences (*p* < 0.05; Tukey’s test).

**Figure 7 animals-14-02516-f007:**
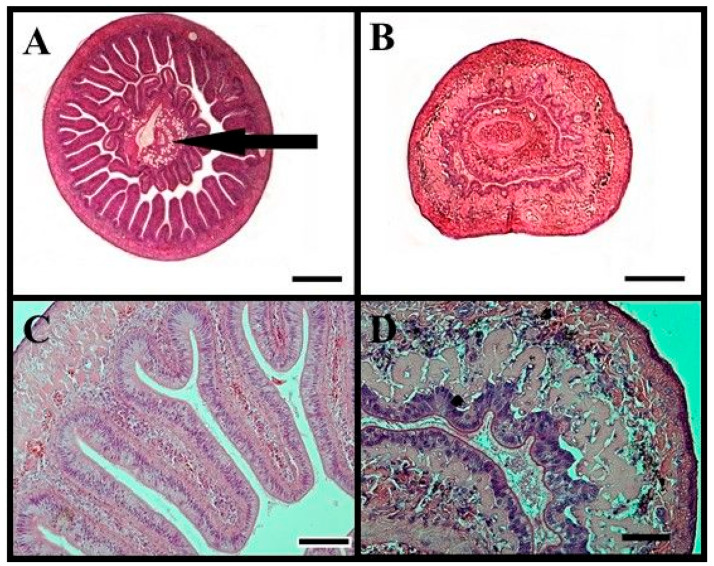
Histological cross-sectional image (H&E) in the midgut of female river lamprey, *Lampetra fluviatilis,* collected in November ((**A**) bar = 500 μm; (**C**) bar = 200 μm) and May ((**B**) bar= 100 μm; (**D**) bar = 200 μm), held under controlled conditions mimicking natural conditions (water temperature and photoperiod) during spawning migration. (**A**,**C**)—transverse image of intestinal diameter, (**B**,**D**)—fragment of intestinal villi (**C**,**D**). The arrow swallows the lipid vacuoles.

**Table 1 animals-14-02516-t001:** Dynamic body weight (g) and length (cm) changes in female river lamprey, *Lampetra fluviatilis,* between November and May (n = 42), held under controlled conditions mimicking natural conditions (water temperature and photoperiod) during spawning migration. The dates in a row are marked with different letters indicate statistically significant differences (*p* < 0.05; Tukey’s test).

Month	NOV	DEC	JAN	FEB	MAR	APR	MAY
Body weight (g)	141.8 ± 10.3 ^a^	126.8 ± 11.7 ^b^	125.6 ± 7.4 ^b^	116.3 ± 5.2 ^bc^	106.0 ± 8.1 ^c^	103.5 ± 6.5 ^c^	100.3 ± 10.5 ^c^
Body length (cm)	43.2 ± 0.1	40.7 ± 1.4	40.9 ± 0.6	39.6 ± 1.2	38.9 ± 1.6	36.3 ± 0.7	35.5 ± 1.7

## Data Availability

Data are contained within the article.

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
