# Peer review of "Histomorphological and Dynamical Changes in Female River Lampreys during Maturation under Controlled Conditions as a Part of Lamprey Restoration Programs"

_animals, 2024, doi:10.3390/ani14172516_

Round 1
Reviewer 1 Report
Comments and Suggestions for Authors
Comments to the Author:
The authors systematacially reviewed histomorphological and dynamical changes in the female river lampreys during maturation. The results will provide a guidance for the restitution of lampreys and their artificial reproduction. However, some suggestions and revisions are needed for the paper before it could be preferably accepted.
1. Please add the growth index, such as body weight and body length, to the results at each sampling time.
2. In line 31-32, “Female river lampreys were kept in conditions that mimicked the natural environment”. Please add the detailed settings of these spawner tanks, including the photo of exterior tanks, and the values of water quality and currents, to “2.3. Broodstock management”.
3. The concentration of reproductive hormone intuitively indicates the reproductive status of broodstock, so I highly suggest to add related dynamical changes in oocyte maturation in this species.
Comments on the Quality of English LanguageMinor editing of English language required.
Author Response
- Please add the growth index, such as body weight and body length, to the results at each sampling time.
Answer: Table with body weight and body length was added to MS.
- In line 31-32, “Female river lampreys were kept in conditions that mimicked the natural environment”. Please add the detailed settings of these spawner tanks, including the photo of exterior tanks, and the values of water quality and currents, to “2.3. Broodstock management”.
Answer: The spawners were kept in tanks operating in a closed aquaculture system (RAS), described earlier by Kujawa et al., (1999). The system consists of 4 pools of 1 m3 capacity, in which the breeders are kept, often separated by sex, and 2 retention tanks, filter. And in devices for regulating temperature (heating or cooling) controlled by a thermoregulator and light control systems connected to a control timer or a photocell led outside the building. Additionally, the pools are covered, preventing the breeders from jumping out. The color of the tangs walls is dark green.
Information was added in MS.
- The concentration of reproductive hormone intuitively indicates the reproductive status of broodstock, so I highly suggest to add related dynamical changes in oocyte maturation in this species.
Answer: Unfortunately, we did not test any hormones in this species at the beginning or end of the experiment. Because lampreys die after a while, unfortunately we no longer have this population in our laboratory. Determining the steroid hormones responsible for oocyte development will only be possible after repetition and further studies. However, we will consider the reviewer's suggestions in future studies on this species.
Reviewer 2 Report
Comments and Suggestions for Authors
PDF is included, attached to this review sheet.

Comments on the Quality of English LanguageThe English is pretty good. Just a few word changes, and remove the anthropogenic possessives for lampreys and fish.
Author Response
Line 29: Delete: „own”
Answer: I was correct.
Line 37: Delete: „and”
Answer: I was correct.
Line 40: Also, there was....
Answer: I was correct.
Line 40: In the oocyte
Answer: I was correct.
Line 71: After reaching only several…
Answer: I was correct.
Line 73: fish; often these species are economically valuable; e.g., sea trout…
Answer: I was correct.
Line 76: undertake a spawning migration will develop.…
Answer: I was correct.
Line 79: but is now regarded…
Answer: I was correct.
Line 82: northwestern
Answer: I was correct.
Line 82: comma, delete parentheses.
Answer: I was correct.
Line 90: spawning act itself
Answer: I was correct.
Line 94: details of the reproductive biology
Answer: I was correct.
Line 94: Delete: in detail
Answer: I was correct.
Line 95: young for stocking.
Answer: I was correct.
Line 96: reproductive biology
Answer: I was correct.
Line 96: Delete „party”
Answer: I was correct.
Line 96: partly under
Answer: I was correct.
Line 100: Delete: also
Answer: I was correct.
Line 107: must
Answer: I was correct.
Line 109: mimic
Answer: I was correct.
Line 112: restoring
Answer: I was correct.
Line 112: have been conducted
Answer: I was correct.
Line 113: Delete „out”
Answer: I was correct.
Line 121: At this time
Answer: I was correct.
Line 122: Add „but”
Answer: I was correct.
Line 122: must be granted for obtaining
Answer: I was correct.
Line 123: Delete: „was obtained”
Answer: I was correct.
Line 124: conducted
Answer: I was correct.
Line 124: in Poland
Answer: I was correct.
Line 129: insert map of collection sites here.
Answer: Due to the large number of figures in the MS, we decided that instead of a map we would add a detailed sampling location in the MS text.
Line 135: 1m3
Answer: I was correct.
Line 149: 100/m3
Answer: I was correct.
Line 149: 100 g/m3
Answer: It was corrected
Line 154: deleted “then”, then
Answer: It was corrected
Line 155: all specimens
Answer: It was corrected
Line 191: largest or biggest
Answer: It was corrected
Line 194: Put the number of specimens used in creating this graph (n = ?).
Also, are the number of specimens examined the same for each month? Last sentence -- put the statistical test used to determine significance.
Answer: It was corrected
Line 195: monthly
Answer: It was corrected
Line 197: All figure captions need to be changed. "Their" refers to lamprey, and they cannot be possessive. Use "the".
Answer: It was corrected
Line 204: Put the number of specimens used in creating this graph (n = ?).
Also, are the number of specimens examined the same for each month? Last sentence -- put the statistical test used to determine significance.
Answer: It was corrected
Line 228: Include the number of specimens (n = ?). Change "their" to "the". Last sentence, indicate statistical test.
Answer: It was corrected.
Line 211: the
Answer: It was corrected
Line 228: Include the number of specimens (n = ?). Change "their" to "the". Last sentence, indicate statistical test.
Answer: It was corrected.
Line 211: the
Answer: It was corrected
Line 228: Include the number of specimens (n = ?). Change "their" to "the". Last sentence, indicate statistical test.
Answer: It was corrected.
Line 257: See above comment : no
Answer: It was corrected.
Line 267: n = ?
statistical test used.
change their to the. It was corrected.
Do all months have the same number of specimens examined? Yes
Answer: It was corrected.
Line 268: natural
Answer: It was corrected.
Line 291: at ages 3 and 4,
Answer: It was corrected.
Line 297: that slow
Answer: It was corrected.
Line 305: However, a greater GSI observed in
Answer: It was corrected.
Line 307: monitoring
Answer: It was corrected.
Line 327: in relation to the mean wet weight of 173.8 g
Answer: It was corrected.
Line 344: April of this study,
Answer: It was corrected.
Line 357: were observed
Answer: It was corrected.
Line 365: of this study,
Answer: It was corrected.
Line 412: on the one hand,
Answer: It was corrected.
Line 438: Check all references for consistency in capitalizing words in the title.
check all scientific names to ensure italics.
N carriage return (enter) after a line preceding the ending -- it puts extra spaces between the words.
Answer: It was corrected.

Reviewer 3 Report
Comments and Suggestions for Authors
Review of “Histomorphological and dynamical changes in the female river lampreys, during maturation under controlled conditions”
This manuscript describes histological changes in stomach, liver, and ovaries during river lamprey migration period. Authors maintained lampreys in captivity and sampled six individuals monthly. The data appear to be valid and robust. I would like to see additional information about condition / weight change of these lampreys. They do not feed during migration, and were not fed in captivity?
Specific suggestions:
Lines 16-19: not all lampreys use oceans. Many US lampreys spend their lives in freshwater only. See your reference 5.
Lines 76-77: modify to “The digestive system of lampreys that undertake spawning migrations atrophies because of a long-term. . . “
Line 79: add “yet” after the ,
Line 192: what was the female weight loss?
Line 201: Figure 1 legend. Change to “Dynamic” changes
Line 205: modify to “. . . with different letters to indicate . . . “
Line 293: “length”
Comments on the Quality of English LanguageSee above
Author Response
This manuscript describes histological changes in stomach, liver, and ovaries during river lamprey migration period. Authors maintained lampreys in captivity and sampled six individuals monthly. The data appear to be valid and robust. I would like to see additional information about condition / weight change of these lampreys. They do not feed during migration, and were not fed in captivity?
Answer: Lampreys, like the European eel, stop eating during migration. Female river lampreys in our study were not fed. A table with the results of the average weight and length of the body weight of female river lampreys was added to MS.
Specific suggestions:
Lines 16-19: not all lampreys use oceans. Many US lampreys spend their lives in freshwater only. See your reference 5.
Answer: I was correct.
Lines 76-77: modify to “The digestive system of lampreys that undertake spawning migrations atrophies because of a long-term. . . “
Answer: I was correct.
Line 79: add “yet” after the,
Answer: This sentence has already been corrected
Line 192: what was the female weight loss?
Answer: Female lampreys lost about 30% of their body weight. This information was added in MS.
Line 201: Figure 1 legend. Change to “Dynamic” changes
Answer: I was correct.
Line 205: modify to “. . . with different letters to indicate . . . “
Answer: I was correct.
Line 293: “length”
Answer: I was correct.

Round 2
Reviewer 1 Report
Comments and Suggestions for Authors
Comments to the Authors:
The results researched the role of the liver in the oocyte maturation in the river lampreys. I suggested that the manuscript could be accepted in present form.
Comments on the Quality of English LanguageMinor editing of English language required.